# Probing into the Effects of Grapevine Leafroll-Associated Viruses on the Physiology, Fruit Quality and Gene Expression of Grapes

**DOI:** 10.3390/v13040593

**Published:** 2021-03-31

**Authors:** Yashu Song, Robert H. Hanner, Baozhong Meng

**Affiliations:** 1Department of Molecular and Cellular Biology, University of Guelph, Guelph, ON N1G 2W1, Canada; yashu@uoguelph.ca; 2Department of Integrative Biology and Biodiversity Institute of Ontario, University of Guelph, Guelph, ON N1G 2W1, Canada; rhanner@uoguelph.ca

**Keywords:** grapevine, grapevine leafroll disease, grapevine leafroll-associated viruses, *Ampelovirus*, *Closterovirus*, *Velarivirus*, *Closteroviridae*, qRT-PCR, microarray, RNA-Seq, transcriptomics

## Abstract

Grapevine leafroll is one of the most widespread and highly destructive grapevine diseases that is responsible for great economic losses to the grape and wine industries throughout the world. Six distinct viruses have been implicated in this disease complex. They belong to three genera, all in the family Closteroviridae. For the sake of convenience, these viruses are named as grapevine leafroll-associated viruses (GLRaV-1, -2, -3, -4, -7, and -13). However, their etiological role in the disease has yet to be established. Furthermore, how infections with each GLRaV induce the characteristic disease symptoms remains unresolved. Here, we first provide a brief overview on each of these GLRaVs with a focus on genome structure, expression strategies and gene functions, where available. We then provide a review on the effects of GLRaV infection on the physiology, fruit quality, fruit chemical composition, and gene expression of grapevine based on the limited information so far reported in the literature. We outline key methodologies that have been used to study how GLRaV infections alter gene expression in the grapevine host at the transcriptomic level. Finally, we present a working model as an initial attempt to explain how infections with GLRaVs lead to the characteristic symptoms of grapevine leafroll disease: leaf discoloration and downward rolling. It is our hope that this review will serve as a starting point for grapevine virology and the related research community to tackle this vastly important and yet virtually uncharted territory in virus-host interactions involving woody and perennial fruit crops.

## 1. The Family Closteroviridae: A Brief Overview

*Closteroviridae* is a family of phloem-limited, positive sense, single-stranded RNA plant viruses that mostly infect woody perennials including important fruit crops such as citrus and grapevine. The family *Closteroviridae* is comprised of four genera: *Ampelovirus*, *Closterovirus*, *Crinivirus* and *Velarivirus*. *Closteroviridae* family members share several conserved features including their long filamentous virion, large genomes with a 5′ cap but no poly-A tail, the presence of a homologue of cellular HSP70, and transmission of all but velariviruses in a semi-persistent manner by various Hemiptera insects that feed on the phloem tissue [1,2]. Vector-mediated transmission of viruses of the *Closteroviridae* is genus-specific. For example, viruses of the genus *Closterovirus* are transmitted by aphids, those of the genus *Ampelovirus* by mealybugs and soft scale insects, whereas viruses of the genus *Crinivirus* by whiteflies [3]. It is unknown if members of the genus *Velarivirus* are transmitted by insects [3].

The *Closteroviridae* family contains some of the largest RNA viruses. In fact, the only family of RNA viruses with larger genomes is *Coronaviridae* [4]. Viruses of the family *Closteroviridae* encode several conserved proteins that are organized in gene blocks each with a distinctive function. ORF1a and ORF1b are translated directly from viral genomic RNA. As members of Alphavirus-like supergroup of (+)ssRNA viruses, closteroviruses use subgenomic RNAs (sgRNAs) as an expression strategy. Genes downstream of ORF1a and ORF1b are expressed through a nested set of sgRNAs that share sequence at their 3′ ends. ORF1a and ORF1b constitute the replication gene block (RGB), the translation products of which are replicative proteins involved in genome replication and transcription. Translation of ORF1a generates a polyprotein with several domains characteristic of the Alphavirus-like supergroup: papain-like leader proteases (L-Pro), RNA methyltransferase (MET), and RNA helicase (HEL) (Figure 1). RdRP is encoded by ORF1b, which is translated via +1 translational frameshift during translation of ORF1a. Ribosomal frameshifting occurs in a small percentage of cases, generating a larger polyprotein with RdRP as the C-terminal extension [3,5]. MET is involved in viral mRNA capping, HEL functions to unwind viral RNA during replication and transcription process, and RdRp catalyzes viral RNA synthesis [1]. The L-Pro domain is a leader protease and is involved in cleavage of the viral polyproteins. In addition, L-Pro plays a crucial role in enhancing viral RNA amplification and long-distance transport [1,6]. *Citrus tristeza virus* (CTV) and *Grapevine leafroll-associated virus 2* (GLRaV-2), both members of the *Closterovirus* genus, encode two copies of the protease domains (L-Pro 1 and L-Pro 2) (Figure 1B). The tandem L-Pro repeats are believed to have evolved from gene duplication, followed by divergence [7]. L-Pro domains in the other three genera of the *Closteroviridae* likely play similar roles although this has yet to be confirmed experimentally.

ORFs 2–6 of members of the *Closteroviridae* (ORFs 3–7 in CTV and GLRaV-3) constitute the quintuple gene block (QGB) and encode proteins involved in virion assembly and intercellular movement (Figure 1). The QGB is another signature conserved across this family of viruses. The QGB encodes five proteins: a small membrane protein, a HSP70 homologue (HSP70h), a protein of about 60 kDa (ranging between 55 and 64 kDa), a capsid protein (CP) and a minor capsid protein (CPm) [8]. The small hydrophobic protein is an integral transmembrane protein that is associated with the ER, which is essential for the local (i.e., cell-to-cell) movement of the virus [9,10]. HSP70h, a homologue of cellular molecular chaperon HSP70, is believed to have been acquired by the ancestor of closteroviruses via RNA recombination [3,11]. HSP70h may play multiple roles in the viral replication cycle, including viral replication, virion assembly as well as cell-to-cell movement through plasmodesmata [3]. CP forms the main body of the virion while CPm, together with HSP70h and p60, form the terminal structure at one end of the virion, giving the appearance of the so called “rattlesnake” morphology characteristic of most members of the *Closteroviridae* [3,12]. It is worth noting that subgroup II viruses of the genus *Ampelovirus,* such as GLRaV-4 and GLRaV-13, lack CPm and their virions do not have such a bipolar structure [13].

Unlike all other viruses in the family *Closteroviridae*, members of the genus *Crinivirus* have either bipartite or tripartite RNA genomes [14,15] and mostly infect herbaceous plants. Viruses of the *Crinivirus* genus also have a small ORF that is located between the ORFs for p60 and CP of the QGB, which is unique to the *Crinivirus* genus (Figure 1D) [15]. While the function of the small protein encoded by this extra ORF remains unknown, it has been shown that p9 of lettuce infectious yellows virus (LIYV) self-interacts [16], which is characteristic of RNA silencing suppressors (RSS) encoded by different viruses [17,18,19]. In line with this, it has been reported that p9 is required for establishing efficient infection and pathogenicity [20]. However, function of p9 and analogous proteins in criniviruses as RSS needs to be experimentally demonstrated.

*Ampelovirus* is the most heterogeneous genus of the family in terms of genome size, architecture and gene content of member viruses [3,21]. For this reason, this genus is further divided into two subgroups [21]. Subgroup I viruses have larger and more complex genomes (larger than 15,000 nt, with 10 or 13 ORFs). Note that there has been inconsistency in the literature regarding whether ORF1a and ORF1b should be counted as one or two separate ORFs. This is largely due to the fact that ORF1b cannot be translated independently but is translated via a ribosomal frameshifting mechanism contingent upon ORF1a translation. To avoid further confusions in the literature, we propose here that ORF1a and ORF1b be considered as two ORFs. In fact, the same numeration system is used for CTV [22], GLRaV-4 [23] and LIYV [24]. In contrast, subgroup II viruses have smaller genomes with fewer ORFs (approximately 13,000–14,000 nt, with 7 ORFs) [21]. As stated earlier, subgroup II viruses lack the ORF for CPm (Figure 1A) [3,21]. It is peculiar that all the GLRaVs of the *Ampelovirus* genus encode a novel AlkB domain in the translation products of ORF1a. Similar AlkB domains are present in many viruses of the family *Betaflexiviridae* that mostly infect woody perennials [25]. Surprisingly, such a domain is not identified in two other viruses associated with GLRD: GLRaV-2 (genus *Closterovirus*) and GLRaV-7 (genus *Velarivirus*). The function of this viral AlkB domain in these viruses remains as an interesting question that needs to be answered. It has been suggested that viral AlkB domains may enhance the long-term survival of these woody plant viruses in their perennial hosts via safe-guarding their genomic RNA against damaging methylation [25,26].

## 2. Grapevine Leafroll Disease and GLRaVs

Among the crop plants susceptible to closteroviruses, grapevine (*Vitis* spp.) is one of the most heavily impacted by infection with GLRaVs. Grapevine is a major fruit crop with global cultivation over 7.4 M hectares. It is of high economic value, being used in the production of table fruit, raisins, juice, seed oil and perhaps most importantly wine [27,28]. Grapevine leafroll disease (GLRD) is economically the most damaging viral disease complex affecting global grape and wine production [29]. So far, six distinct species of viruses belonging to three genera of the family *Closteroviridae* are reportedly associated with GLRD. These viruses are named grapevine leafroll-associated virus 1, 2, 3, 4, 7, and 13 (GLRaV-1, -2, -3, -4, -7 and -13). GLRaV-2 is a member of the genus *Closterovirus*; GLRaV-7 is the prototype member of the genus *Velarivirus*, while the other viruses belong to the genus *Ampelovirus*. It is important to note that none of these viruses has been proven as the causal agent of GLRD.

GLRD has a worldwide prevalence and is found in almost all regions where commercial grapevines are grown [30]. The main transmission route of GLRaVs is through global exchange and propagation of virus-infected grapevine material, which resulted in the current worldwide distribution of GLRD [31]. GLRaVs can also be transmitted to adjacent vines in the same vineyard and to nearby vineyards via phloem-feeding mealybugs and scale insects in a semi-persistent manner (Table 1) [31]. In this mode of transmission, viruses are acquired by insects via sap-feeding and are retained for some time within insects without replicating. Viruses are transmitted to new hosts when insects probe and feed again on other vines [32,33]. Vineyards affected with GLRD can suffer yield reduction from 30% to 50% with an altered fruit chemistry that negatively impacts the quality of berries, juice and wine. Infections with GLRD shorten the production lifespan of a vineyard. Typical symptoms of infection include downward curling of leaf margins of mature leaves, red to purple discoloration of leaves of dark-berried cultivars, and chlorotic discoloration in white-berried cultivars (Figure 2) [29]. GLRaV-3 is considered the major agent for GLRD. It is estimated that infection by GLRaV-3 alone may lead to an economic loss of between $25,000 and $41,000 per hectare over the lifespan of a vineyard [34,35]. The only strategy available to eradicate GLRaVs from the grapevine host is the elimination of viruses through micro-shoot tip tissue culture, thereby generating virus-free stocks. Currently, the main strategies to control and manage GLRD spread is through propagation of clean stock and biological vector control [30].

### 2.1. Multiple Viruses of the Genus Ampelovirus Infect Grapevine

To date, four distinct species of viruses of the genus *Ampelovirus* have been identified in grapevine. GLRaV-1 and GLRaV-3 are more prevalent and commonly associated with GLRD, while GLRaV-4 and GLRaV-13 are much less common and their role in GLRD remain questionable.

GLRaV-1 belongs to subgroup I of the genus *Ampelovirus*. The complete genome sequence of two GLRaV-1 isolates (WA-CH and WA-PN) revealed its genome size of 18.7–18.9 kb with 10 ORFs (Figure 1A) [37], making GLRaV-1 the plant virus with the second largest RNA genome, only after CTV. GLRaV-1 is the only member of the family that encodes two CPm copies—CPm1 and CPm2 [21,37]. The function of these duplicated CPms remains unknown, though it is speculated to have resulted from gene duplication at some point over the evolution of closteroviruses [38]. As a viral species, GLRaV-1 has a high degree of variability when compared to other GLRaVs [39,40,41]. Sequence identities among GLRaV-1 variants can be as low as 59.5% in certain regions of the genome [39], with CPm2 being the most variable, followed by CPm1 and HSP70h [39,40]. Such hyper sequence variations in certain ORFs occurs likely because they play non-essential roles in viral survival, thereby being subjected to no or less selection pressure [39,40].

The two ORFs located at the 3′ proximal region of the viral genome are unique to GLRaV-1 [37]. The sgRNA encoding p24 was shown to be expressed at a much higher level than any other sgRNA including those for CP and p21 [37]. This p24 was recently identified as a RSS [19]. Plants utilize RNA silencing as a defense mechanism against viral infections that is triggered by dsRNA-intermediates generated during viral replication [42]. Viral RSSs counteract host defense responses and function as viral pathogenicity determinants leading to the development of disease symptoms of the host upon infection [43,44]. p24 of GLRaV-1 accumulates as dimers in the nucleus and exhibits RSS activity that interferes with both the local and systemic host RNA silencing when assayed using co-infiltration of transgenic *Nicotiana benthamiana* 16c expressing GFP. In addition, p24 elicits local and systemic necrosis that resembles a hypersensitive response [19].

GLRaV-3 is regarded as the major inducer of GLRD and is the most destructive virus of grapevine due to its global distribution and high economic impact. This virus is found at a significantly higher prevalence than any other GLRaVs worldwide and is most frequently associated with severe GLRD symptoms, negative impacts on grapevine yield, fruit quality, and the production lifespan of vineyards [35,45,46,47,48]. For this reason, GLRaV-3 has been the focal point of research pertaining to its effects on grape physiology, quality and yield as well as virus-host interactions.

GLRaV-3 is the prototype member of the genus *Ampelovirus*. It has a monopartite genome of 18.4–18.6 kb with 13 ORFs (Figure 1A). The GLRaV-3 genome contains the RGB and QGB typical of family *Closteroviridae*. Like several other viruses of the *Ampelovirus*, GLRaV-3 encodes an AlkB domain as part of the translation product of ORF1a. Interestingly, there is a small ORF (i.e., ORF2) between the RGB and QGB. ORF2 potentially encodes a 6 kDa protein (p6). It remains questionable however if ORF2 is actually translated, let alone if its translation product has a function essential for GLRaV-3 replication and infection. ORF2 is absent in all GLRaV-3 isolates of phylogenetic group VI [49,50,51,52,53].

The remaining ORFs, ORF8-ORF12, are unique to GLRaV-3 (Figure 1A). Whereas the functional roles of these ORFs have yet to be elucidated, work is in progress to characterize their functions. When examined using the co-infiltration assay involving transgenic *N. benthamiana* 16c plants, p20B encoded by ORF10 was shown to function as a RSS, whereas p21 (ORF8) and p20A (ORF9) do not [54]. Interestingly, sgRNA for p20B is the most abundant in infected tissue, followed by sgRNAs for p20A, p21 and CP [55]. Abundant expression of sgRNAs corresponding to RSS is also observed in other viruses of the family *Closteroviridae*, including GLRaV-1 (p24), GLRaV-2 (p24), CTV (p23) and BYV (p21) [37,56,57]. Another feature is that each gene encoding an RSS in these viruses is located at the 3′ most proximal position of their genomes. One exception to this is GLRaV-3 in which two additional small ORFs are downstream of ORF10, the gene encoding RSS. These two ORFs (ORF11 and ORF12) are highly variable both in sequence and in length among GLRaV-3 variants. No analogous ORFs exist for any other viruses of the family. Based on these observations and as suggested by Maree et al., 2013, ORF11 and ORF12 may not encode proteins [48], or their translation products may be dispensable for GLRaV-3. Further work is required to ascertain the authenticity of ORF11 and ORF12.

GLRaV-13 is the newest addition to the *Ampelovirus* genus [58]. It was identified in a Japanese wine grape cv. Koshu (*V. vinifera*) grafted onto the rootstock Kober 5BB (Dr. Ito, personal communication). The source vine, a177, exhibited typical GLRD symptoms. GLRaV-13 has an RNA genome of 17,608 nts with a genome structure that is similar to GLRaV-1 and GLRaV-3. Its 5′ UTR is 1100 nts long, which is longer than those in GLRaV-3 isolates (510–802 nts) and GLRaV-1 (781 nts). Phylogenetic analyses using HSP70h, RdRP and CP showed that GLRaV-13 is a member of subgroup I of the *Ampelovirus* genus. Besides the RGB and QGB common to all members of the *Closteroviridae* family, GLRaV-13 contains four ORFs in its unique region near the 3′ end of the genome, potentially encoding p22, p7, p8 and p23 [58]. Because the source vine was co-infected with GLRaV-3, it remains questionable if GLRaV-13 alone can induce leafroll symptom.

GLRaV-4 belongs to subgroup II of the genus *Ampelovirus*. Compared to GLRaV-1 and GLRaV-3 (subgroup I), GLRaV-4 has much a smaller genome (ca 13,830 nt) encoding 7 ORFs (Figure 1A) [59]. Since its identification in 1990 [60], a large number of genetically diverse GLRaV-4 strains have been reported, including strain 5, 6, 9, De, Car, Pr, Ru1, Ru2, and Ob [59,61,62,63,64,65,66]. These strains were originally described as distinct GLRaV species based on their seemingly lack of serological relatedness [21,23]. Later on, phylogenetic analyses of the taxonomically relevant genes (HSP70h, CP and RdRP) among these viruses revealed that they all had levels of amino acid sequence divergence below the 25% threshold as defined by the International Committee on the Taxonomy of Viruses (ICTV) for the demarcation of individual species for the family *Closteroviridae* [13,21]. Considering the fact that these viruses share identical genome structure, similar genome size, as well as similar biological traits, they were later re-classified as individual strains of GLRaV-4 [21,23].

### 2.2. GLRaV-2, a Member of the Genus Closterovirus

GLRaV-2, a member of the genus *Closterovirus,* has a monopartite RNA genome of 16.5kb with 9 ORFs [67] (Figure 1B). While members of *Closterovirus* such as CTV and BYV are transmitted by aphids [3], no known biological vector has been identified for GLRaV-2 (Table 1) [67]. Besides its association with GLRD, GLRaV-2 is also implicated in graft incompatibility and rootstock stem lesion [68,69,70,71]. GLRaV-2 strain PN is associated with both GLRD and graft incompatibility, while strain RG is associated with severe graft incompatibility but does not induce GLRD symptoms [71]. Yet another isolate of GLRaV-2, strain SG causes asymptomatic infections with little impact on either yield or fruit quality [72].

The last two ORFs of GLRaV-2 encoding p19 and p24 are unique to the virus (Figure 1B). GLRaV-2 p24 is a functional homologue of BYV p21 and CTV p20. Each of these proteins encoded by the 3′ terminal ORF in their corresponding viruses has been demonstrated to function as a RSS [73,74,75]. GLRaV-2 p24 induces systemic necrosis in *N. benthamiana* when expressed via a potato virus X (PVX) vector and local necrosis via a vector based on barley stripe mosaic virus (BSMV) [18]. A recent study suggests that p24 may function by directly binding siRNAs thereby preventing their incorporation into RNA-induced silencing complex (RISC) [17]. These results are indicative of the roles of p24 in GLRaV-2 pathogenesis and symptom development. It is tempting to suggest that the properties of p24 encoded by different strains are related to and even responsible for the different diseases associated with GLRaV-2. On the other hand, the function of p19 has yet to be elucidated. Although GLRaV-2 has a genomic structure virtually identical to BYV and GLRaV-2 p19 is similar in size and genomic location to BYV p20 [76], only limited sequence similarity exists between these two proteins [76,77]. The possibility that GLRaV-2 p19 may function in systemic movement or as a component of the virion tail, as demonstrated for BYV p20 [11,78], awaits experimental validation.

### 2.3. GLRaV-7, the Prototype Member of the Genus Velarivirus

Before the identification and genome sequencing of GLRaV-7, only three genera were recognized for the family *Closteroviridae*. In 2012, two independent research groups published the complete genome sequence of two isolates of GLRaV-7. Isolate PN-23 was from a vine of cv. Pinot noir (clone 23) that exhibited no symptoms of GLRD [79]. The other isolate, designated AA42, was from an unidentified symptomless white wine grape cultivar from Albania. When graft-inoculated onto Cabernet franc, a sensitive indicator for GLRD, AA42 induced mild leafroll symptoms [80]. Based on multiple differences between GLRaV-7 and other viruses of the family, it was proposed that a new genus be established to contain GLRaV-7, little cherry virus 1 (LChV-1) and cordyline virus 1 (CoV-1) [79,80]. As the prototype member of the newly established genus *Velarivirus*, GLRaV-7 has been detected in multiple countries [81,82,83,84]. Unlike other viruses of the *Closteroviridae*, no insect vector has been found to transmit GLRaV-7 (Table 1) [21]. Its distribution is therefore attributed to the global exchange of propagating materials and vegetative propagation [85]. Isolates of GLRaV-7 have genome size of 16.4–16.5 kb with 9 (for isolate PN-23) or 10 (for isolate AA42) ORFs (Figure 1C) [3,79,80]. Isolate AA42 contains an extra ORF between those encoding HSP70h and p61, which potentially encodes a protein of 10 kDa resembling the extra ORFs found at similar genome positions in CoV-1 but not in LChV-1. Interestingly, a similar ORF is present in RNA2 of criniviruses [3,80]. The function of p10 remains unclear.

## 3. Effects of GLRaVs on Physiology, Fruit Quality and Gene Expression

Few studies have been conducted to understand the molecular interaction between GLRaV-2, -4, -7 and their grapevine host. Unlike GLRaV-1 and GLRaV-3 that induce strong GLRD symptoms, GLRaV-2, -4 and -7 are associated with asymptomatic or mild GLRD symptoms with apparently little impact on either fruit yield or quality [59,63,66,72,79,86]. GLRaV-2 has an erratic variance in its pathological properties and associated diseases. Infection with GLRaV-7 alone does not induce GLRD symptoms, and it is questionable if it should even be called a GLRaV [66,79,86]. In this section, we will focus our discussion on the effects of GLRaV-1 and GLRaV-3 infections on the grapevine host.

GLRaV-1 is the second most prevalent virus that is associated with GLRD after GLRaV-3 [31]; however, studies on its virus-host interaction are still in their infancy. The few studies that have been conducted in an attempt to reveal the physiological impacts on grapevines reported inconsistent and sometimes conflicting results. For example, berries of cv. Nebbiolo vines mixed-infected with GLRaV-1, GVA and GRSPaV were reported to have decreased bud burst index but have heavier berry weight, higher acidity and resveratrol content. No changes in either yield or the contents of soluble solids and phenolic compounds (anthocyanins and catechins) were observed [87]. In contrast, Santini et al., 2011 reported an increase in berry weight and titratable acidity but a reduction in yield in Nebbiolo vines co-infected with GLRaV-1 and GVA [88]. On the other hand, Guidoni et al., 2000 reported no change in yield, but documented decreased berry weight and titratable acidity level in vines co-infected with GLRaV-1 and GVA [89]. Ghaffari et al., 2020 reported that neither yield nor berry photosynthetic rate was affected by GLRaV-1 infection. In addition, they found that Pinot noir vines infected with GLRaV-1 alone had increased sugar concentration [90].

The effect of GLRaV-1 infection at the transcriptomic level on berries of Pinot noir (clone RAC 68) was recently investigated at veraison [90]. Veraison is a developmental stage of grapevine characterized by the onset of berry ripening when 50% of the berries have changed color [91]. No significant difference in gene expression was found for genes involved in flavonoids pathways, including biosynthesis of anthocyanin, pro-anthocyanidins and flavonol [90]. Combined with findings from Giribaldi et al. (2011), where mixed infection involving GLRaV-1 did not alter anthocyanin accumulation in berries [87], GLRaV-1 infection alone may have no or minimal impact on flavonoids biosynthesis in berries. The expression of hexose transporter 1 (HT1) was significantly down-regulated in berries of vines infected with GLRaV-1 at veraison [90]. A range of genes involved in host stress response and defense were significantly differentially regulated. Proteins in the heat shock family, as well as heat shock transcription factors, were significantly upregulated. The disease resistance (R) gene encoding the protein MLA was downregulated. R proteins are produced by the plant host to recognize effectors generated by pathogens, thereby activating effector-triggered immunity (ETI) against infection [92]. AGO2 and DICER 2 involved in RNA silencing were also up-regulated (Table S7 in [90]). Whether these genes are involved in a general response against environmental changes/pathogen attack or in a virus-specific response to GLRaV-1 infection remains undetermined.

Grapevines infected with GLRaV-3 generally do not show GLRD symptoms until veraison [93]. Numerous investigations have been conducted to reveal the effects of GLRaV-3 infection on the physiology and enological properties of infected grapevines. Similar to the situation with GLRaV-1, these studies reported highly variable findings, which can be difficult to interpret and confusing at times. For example, several studies reported that GLRaV-3 infection reduced berry weight [89,94,95,96], while others claimed that infection with GLRaV-3 either had no effects or actually increased berry weight [88,97,98,99,100,101]. Many factors may be attributable for the discrepancy between these studies, including differences in the cultivars and rootstocks used, the genetic variants of GLRaV-3 involved, co-infection with different viruses, age of the infected vines, climate, and methods of sampling. Here we discuss some of the major impacts of GLRaV-3 infection that were reported by the majority of studies.

Berries of GLRaV-3-infected vines of both white and dark-berried cultivars were consistently found to have lower soluble solids (°Brix) and higher titratable acidity [89,94,96,98,100,102,103]. In dark-berried cultivars, GLRaV-3 infection caused decreased anthocyanin and proanthocyanidins contents [89,97,98]. On the other hand, leaves of vines infected with GLRaV-3 were reported to have reduced chlorophyll and carotenoid content, as well as lower rates of stomatal conductance and net CO_2_ assimilation, leading to reduction in photosynthetic efficiency [96,101,102,104,105,106,107,108,109]. There was also a reduction in the level of soluble proteins in GLRaV-3-infected leaves, suggesting an inhibition in protein synthesis that likely contributed to reduced Ribulose-1,5-bisphosphate carboxylase-oxygenase (RuBisCO) and nitrate reductase activities [105,106,110].

Interestingly, a higher level of soluble sugars and starch was reported in leaves of cv. Merlot infected with GLRaV-3 [109]. This correlated well with decreased sugar accumulation in berries [101,103,111]. It can be speculated that sugar translocation from source (leaves) to sink (berries) was restricted due to GLRaV-3 infection. Significant increases of major flavonoids were reported in symptomatic grapevine leaves of cv. Merlot infected with GLRaV-3, including the de novo synthesis of anthocyanins and the increased accumulation of flavonols and proanthocyanidins [112]. In accordance with this quantified increase, earlier appearance of color change due to anthocyanins was observed in leaves of GLRaV-3-infected vines when compared to healthy controls [89]. It is worth noting that in the study by Moutinho-Pereira et al. (2012) on red wine grape cv. Touriga Nacional coinfected with GLRaV-3 and GLRaV-1, lower contents of soluble sugar and protein were detected in leaves. In contrast, the same leaves had higher starch content [106]. Halldorson & Keller (2018) reported that a higher level of soluble sugar was detected in GLRaV-3-infected leaf [109]. The results on sugar accumulation in GLRaV-3-infected leaves found by Moutinho-Pereira et al. [106] contrast with those reported by Halldorson & Keller [109]. Such discrepancy again may be due to multiple factors such as the differences in the infection status of the experimental materials among others. Diverse interactions between the host and its pathobiome may have influenced the outcomes of their research.

Most of the prior studies focused on physiological changes related to vine performance and berry quality due to infection by GLRaV-3. The molecular mechanisms underlying pathological impacts of GLRaV-3 on grapevine have yet to be fully elucidated. Further work is required to understand the molecular interactions between GLRaV-3 and the grapevine host to clarify the complex pathobiology of GLRD.

Sugar transporter genes were up-regulated in GLRaV-3-infected leaves but down-regulated in berries at veraison and harvest [111,113], providing an explanation for the increased sugar accumulation in leaves and the reversal in ripening berry [101,103,109]. The most recent transcriptomic study of berries collected at veraison (seeds excluded) of cv. Pinot noir (clone RAC 68) infected with only GLRaV-1 or co-infected with both GLRaV-1 and GLRaV-3 proposed the use of individual berries to avoid bias introduced by variations between berries [90]. In this study, berries were individually sampled and grouped according to their developmental stages. They were judged by organic acid and sugar levels followed by analysis via downstream transcriptomic profiling. The authors proposed that single infection with GLRaV-1 or co-infection with both GLRaV-1 and GLRaV-3 did not directly lead to the downregulation of flavonoid biosynthesis nor to sugar accumulation in berries as no significant difference was found between virus-infected and healthy control vines. Instead, GLRD indirectly caused reduced sugar accumulation and anthocyanin synthesis, as reported in past studies, through causing a delay in the berry ripening process [90]. While this study provides novel insights in GLRaVs-host interaction, further analysis employing different cultivars and validation assays (qRT-PCR, proteomics and metabolomics) should be conducted for affirmation.

GLRaV-3 infection led to the over-expression of genes involved in the flavonoids biosynthetic pathway in leaves of red wine grape cultivars. Consequently, various products of this pathway such as anthocyanins, proanthocyanidins, and flavonols also experienced a significant increase [112]. In particular, there was de novo synthesis of anthocyanins in GLRD-symptomatic leaves [112]. In contrast, genes involved in the flavonoid biosynthetic pathway were down-regulated in ripening berries of red-skinned grapevine [111], resulting in decreased levels of anthocyanins, proanthocyanidins and flavonols [89,97,98,111]. Interestingly, flavonol accumulation was higher in berries of vines infected with GLRaV-3 than in GLRaV-3-free vines at both pre-veraison and veraison [111]. Flavonol levels declined quickly only after veraison, leading to lower accumulation in GLRaV-3-infected berries at harvest [111].

In studies involving leaves of GLRaV-3-infected vines of cv. Cabernet Sauvignon and Carménère, genes involved in a spectrum of biological functions were differentially regulated. These include the biosynthesis of primary and secondary metabolites, translation, protein processing, hormone metabolism and transport, and defense [113,114]. In particular, genes for chlorophyll biosynthesis enzymes and other proteins involved in photosynthesis were down-regulated [113], in agreement with the repressed photosynthetic activity and decreased chlorophyll accumulation in GLRaV-3-infected leaves as reported by others [96,101,102,104,105,106,107,108,109]. In contrast, the gene for beta-1,3-glucan involved in callose synthesis and genes involved in reactive oxygen species (ROS) scavenging were up-regulated [113,114]. GLRaV-3 infection also led to increased production of ROS, increased accumulation of soluble proteins and free proline in grapevine plantlets [110]. Together, these findings suggested an activation of basal defense response to GLRaV-3 infection, including blockage of plasmodesmata through callose deposition, production of signaling molecules and hormones (SA, JA, ET), oxidative burst and ROS scavenging molecules (e.g., proline).

Small RNAs (sRNAs) were also found to be differentially regulated in leaves of vines infected by GLRaV-3 [115]. These sRNAs were GLRaV-3-specific, suggesting that there was activation of host RNA-silencing as a defense response against GLRaV-3 infection. In addition to sRNAs that are specific to GLRaV-3, multiple microRNAs (vvi-miRNAs) involved in various developmental processes were differentially regulated in both leaf and phloem tissues of infected vines as compared to healthy vines [115,116]. Their possible roles in interaction with GLRaV-3 and contribution to symptom development are topics for future studies.

## 4. Methodologies to Study Effects of GLRaVs on Grapevine

Various methods have been used to unravel the effects of infection by GLRaVs on the grapevine host. The utilization of these methods largely coincided with the availability of technologies. The more commonly used methods are aimed at transcription (quantitative RT-PCR, microarray and RNA-seq). In addition, a few studies attempted the use of a metabolomics-based approach. Here, we focus our discussion on these methods involving transcription, highlighting the strength and limitations of each method. Because of the scarcity of information obtained through metabolomic and proteomic approaches, we will not cover them in this review.

**qRT-PCR:** Relative qRT-PCR is a common method used to monitor changes in the expression of specific genes caused by virus-host interaction. Though qRT-PCR has been broadly used to study changes in gene expression, this method has only been used in few studies involving the effects of GLRaV infections on grapevine. Gutha et al., 2010 first used qRT-PCR to examine changes in the expression of genes of the flavonoid biosynthetic pathways in GLRaV-3-infected leaves. They discovered that a majority of these genes were up-regulated [112]. Quantification of gene expression using qRT-PCR relies on gene-specific primers; as such, only a small number of genes were examined at a time. Despite its non-inclusive nature, qRT-PCR is still favored for its specificity and sensitivity and is often used to validate results derived from transcriptomic studies such as microarray and RNA-Seq.

A crucially important aspect to ensure the validity of results from qRT-PCR is the selection of reference genes. Relative qRT-PCR requires the use of reference genes, whose expression is stable among different tissues, different developmental stages and experimental conditions. Selection of references genes vary greatly from study to study involving different plant species as there is not a set of universally stable genes across plant species and experimental conditions [117]. The choice of reference genes may greatly affect the conclusions derived from qRT-PCR on host response to viral infections. Commonly used reference genes include actin, glyceraldehyde 3-phosphate dehydrogenase (GAPDH), eukaryotic translation elongation factor 1 alpha (eEF-1α), NADH-ubiquinone oxidoreductase chain 5 (NAD5), SAND, and ubiquitin [112,118,119,120]. Studies involving grapevine often rely on the published literature for the selection of candidate refence genes. However, there lacks definitive evidence indicating that any of these reference genes identified in other plant species was also the most stable in grapevines. In addition, few studies were conducted to identify reference genes for differential gene expression analysis among the different grapevine tissues/organs that were infected with GLRaV-3 [36].

Prior studies have also employed data generated from microarray and RNA-Seq to identify the most stable genes in leaf and berry tissues of healthy grapevine [121], grapevine leaf under water and/or heat stress [117], or berries of table grapes at different developmental stages and under different treatments [122]. Unfortunately, there was no consensus among these studies as to which genes are the most stable. RNA-Seq analysis serves as a highly effective approach for the discriminative identification of reference genes to be used in qRT-PCR for specific types of organisms and experiments. However, no such studies have been conducted to identify the most suitable reference genes for the different tissues of grapevines infected with GLRaVs. Through analysis of RNA-Seq data, we have identified three most stable genes (CYSP, NDUFS8, YLS8) for use in qRT-PCR to study the effects of GLRaV-3 infection on grape gene expression.

**Microarray:** Microarray involves hybridization between sequence-specific probes and sample cDNAs to determine relative abundance of specific gene transcripts [123]. While microarray is reliable [124], this system is less desirable and limited in scope because only annotated genes can be detected by this method. Plant species whose genomes are not available or not well annotated are therefore at an inevitable disadvantage. Although gene annotation databases have been refined over the years, there are still a substantial number of unknown genes that are present in the databases, making it difficult to interpretate the data from microarray or RNA-Seq. This technique has been used at the earliest stage of transcriptomic profiling of GLRaV-3-grapevine interaction by Espinoza et al., 2007 [113,114]. Results of their studies have been discussed in the earlier section. The interpretation of their results was largely limited by the state of gene annotation available at the time. Many of the genes whose expression was altered due to GLRaV-3 infection were not annotated [113,114]. Nevertheless, more standardized analytic protocols have been established for microarray assays when compared to RNA-Seq. Because of these limitations, microarray has been largely replaced by RNA-Seq. However, microarray remains as an effective technique for use in studies involving model organisms that have comprehensive annotation profiles of genomes.

**RNA-Seq:** Over the past decade, RNA-Seq has become the method of choice for transcriptomic analysis. One most obvious advantage of RNA-Seq over microarray is its unbiased, all-encompassing nature in detecting transcripts corresponding to both annotated and unannotated genes with genome-wide coverage. This is because RNA-Seq does not require the use of probes or primers specific for certain annotated genes. In addition, RNA-Seq offers absolute gene expression values that increases data resolution. In microarray assays of GLRaV-3-infected grapevine leaf, the detection rate was around 58–72% [113]. Such a rate further dropped when the *Arabidopsis* genome sequence was used for chip design due to sequence differences between the genomes of Arabidopsis and grapevine [114]. RNA-Seq analysis could detect transcripts corresponding to over 90% of the grapevine transcriptome as examined recently in our lab (unpublished data), thus significantly increasing the likelihood of detecting rare transcripts. In another study on the differential gene expression of GRBaV-infected grapevine, detection rate of RNA-Seq also reached up to 87% [125].

Several RNA-Seq platforms are available with different sequencing methods, including sequencing by synthesis, Ion Torrent sequencing, pyrosequencing, and sequencing by ligation. Despite variations in the principles and techniques used by the different platforms, the general workflow remains similar. A cDNA library is constructed using mRNA or rRNA-depleted total RNA, fragmented, followed by attachment of adaptors on both ends of the cDNA for downstream parallel sequencing. Hundreds of thousands to millions of sequencing reads are then produced with varied read lengths depending on sequencing platforms [126]. The resulting sequencing reads constitute the ‘raw data’ of RNA-Seq and are subjected to data processing.

RNA-Seq can be applied to a wide range of analyses, including the expression profiling of both coding and non-coding RNAs, SNP detection, profiling of complete virome and pathogen detection in a host, and de novo genome assembly. Pipelines used to analyze the RNA-Seq data vary depending on the purpose of the research. Here, we focus our discussion on the key steps of the bioinformatic pipelines involved in differential gene expression analysis and the potential pitfalls associated with each step. We also provide what we feel as the best tools for each step based on our own experience. Challenges in using RNA-Seq data to identify genes whose expression is altered due to GLRaV infection stem from difficulties in the interpretation of massive numbers of differentially expressed genes (DEGs) and highly unstandardized analytic protocols for RNA-Seq analysis. A typical workflow for DEG analysis involving an RNA-Seq dataset where the host genome sequence is available includes the following steps: (i) quality control of raw data; (ii) alignment of sequence reads; (iii) feature counts and counts normalization; (iv) differential gene analysis; and (v) gene set enrichment analysis and functional annotation (Figure 3). At each step of the pipeline, various resources are available for researchers, which could lead to bias in the end results if improper choices of tools were made. Below we discuss each of these steps and the technical challenges associated with each step.

**Quality control.** Quality control of raw RNA-Seq data involves the removal of adaptor sequences and low-quality reads. Low-quality reads are defined by their higher probability of incorrect calling and could influence downstream analysis if not removed. On the other hand, use of overly stringent trimming parameters would affect accuracy of gene expression assessment [127]. While it is recommended to trim raw data prior to sequence alignment, there is no consensus as to which tool or parameters will generate the best results. Different trimming tools and parameters have been used by different researchers, which could generate incomparable or even invalid results. The choice of trimming tools should be considered based on the level of refinement of the tool, its update frequency, and its compatibility with the downstream analysis required for the purpose of the study. Currently, some of the most commonly used bioinformatics tool for quality control are TrimGalore and FastQC [128,129]. We recommend that moderate trimming be applied to the raw data generated with RNA-seq.

**Alignment of sequence reads.** RNA-Seq read length varies between sequencing platforms. In general, the longer the reads, the more accurate the assembled transcripts [130]. Therefore, a longer read length is desired for purposes such as de novo assembly and SNP detection. For the purpose of DEGs analysis with known reference genome, Illumina sequencing platforms are most commonly used, which generate read length between 75 and 150 bp. For DEG analysis with known reference genome, the sequence reads are mapped to the reference genome using bioinformatic tools. Again, a large collection of tools is available for alignment of sequence reads and the choice of tools markedly impacts the results generated. Popular conventional alignment tools, such as Bowtie [131] and BWA [132], are still the preferred methods in transcriptomic analysis involving RNA-Seq [133,134,135,136,137]. One drawback of these tools is their inability to detect alternatively spliced variants; therefore they are not advised for use in alignment to a reference genome [130]. Other popular aligner tools, such as STAR [138], GSNAP [139], GSTRUCT, and MapSplice [140], were more favored when evaluated in terms of performance-associated benchmarks such as alignment yield, base-wise accuracy, mismatch, gap placement, and exon junction discovery [141]. Computational resources are another factor to consider when choosing appropriate aligner tools. Alignment of sequence reads against a genome is computationally very expensive because the tools associated with this step require high computational capacity. For instance, the aligner STAR requires on average 30 GB of memory. Other aligners, such as BBMap, takes as much memory as a computer can provide, which could hit around 90 GB [142]. For this reason, cloud computing is required because personal computers are inadequate for such a job. Even with cloud computing, it is still a technically challenge to preform RNA-Seq data analysis as only limited quota is provided to a researcher at a given time. Other factors to consider include the level of refinement of the tools, and their compatibility with other tools that are required for both upstream and downstream steps. 

**Read counts and normalization.** Gene expression is quantified by counting the mapped reads. Quantification of the number of reads mapped to each gene are “raw counts” which cannot be used directly to compare expression between genes or between treatments. These raw counts do not account for within- or between-sample effects such as read length, sequencing depth, and biological conditions. These variations need to be addressed first otherwise raw counts cannot be used to accurately determine gene expression difference. Therefore, the fourth step is normalization of raw counts. The normalization step is an essential and yet another highly variable step in the bioinformatic workflow. Many normalization software are available to address sample effects and all are based on different assumptions and algorithms [143]. Failure in choosing an appropriate normalization method can lead to inaccurate identification of DEGs. Several studies have been conducted to compare the performance of different normalization algorithms. While there is no consensus on what would be the best normalization method, it is generally agreed that RPKM (Reads Per Kilobase per Million mapped reads) and its derivative FPKM (Fragments Per Kilobase per Million mapped reads) performed poorly whereas Med (Median) by DESeq and TMM (Mean of M values) by edgeR give better results [144,145,146,147]. These studies provided preliminary framework for researchers in selecting normalization methods for use in RNA-Seq analysis. For further detail on normalization methods and their underlying assumptions, the reader is referred to a comprehensive review by Evans et al., 2018 [143].

**Identification of DEGs.** After normalization, the sequence data is then subjected to analysis to identify genes whose expression is significantly altered. Again, this step can vary depending on the set threshold of fold-change (FC) of DEGs. FC represents the ratio of expression levels between an experimental sample and the control sample and is calculated using normalized counts by log_2_ (experimental/control). In essence, FC values indicate the degree of differences in the expression of a gene between an experimental sample and a control sample. A positive FC value represents an up-regulation whereas a negative FC value means that a gene is down-regulated. The assignment of a DEG must be based on statistical significance tests. In other words, it cannot be based on an arbitrary FC value. Unfortunately, many researchers often make intuition-based arbitrary choices by using an absolute FC value between 1 and 2 as the cutoff to designate if the expression of a gene is differentially regulated. DEGs with a FC value that is lower than the arbitrary cutoff were considered non-differentially expressed and therefore discarded [133,135,137,148,149,150,151,152]. There is no clear justification as to why such a FC threshold was commonly used. A case of FC of any positive value implies an up-regulated DEG while a negative FC value signals a down-regulated DEG as long as they pass the statistical analysis. Potential DEGs should be considered rather than simply discarded as long as they pass the determined threshold from statistical significance test. When identifying DEGs, one needs to consider the complex and intricate interplay between gene networks and the possibility that a DEG with a small FC value may exert a profound influence on the expression of other genes. As such, stringent cut-off of DEGs may result in the omission of DEGs with a small FC value, leading to incomplete profiling of gene expression.

The outcome of gene set enrichment analysis largely depends on how complete the genome annotation database is for the organism in question. Because this step is beyond the control of transcriptome analysis, it is not discussed here.

Based on the above discussions, it is evident that there is no standard bioinformatic pipeline to use for different research purposes. Our primary purpose is to provide a guideline for researchers facing these technical challenges when conducting virus-host interaction studies using transcriptomics analyses of meta data set generated through RNA-Seq. To serve as a starting point, we describe below a bioinformatics pipeline we recently developed through analysis of RNA-Seq data to identify grapevine genes, whose expression is altered due to infection by GLRaV-3. The bioinformatic workflow is summarized in Figure 3. Raw RNA-Seq data is trimmed by using TrimGalore! using default settings [128]. Subsequently, the quality of data is assessed by FastQC [129]. Trimmed RNA-Seq data is mapped to the grapevine genome (cv. PN40024) by using STAR [138]. The number of reads mapped to each gene feature is then counted by HTSeq Counts [153], which generates a list of protein-coding gene names (Ensemble plant gene ID) with their corresponding read counts in each sample. Normalization of raw read counts and DEGs analysis were conducted by using DESeq2 [154]. False discovery rate (FDR) of 0.05 was used to filter out genes not considered as differentially expressed. In our data analysis, no arbitrary FC threshold was applied. Gene set enrichment analysis and annotation was conducted using PANTHER [155] or DAVID [156].

## 5. Conclusions and Future Research

Grapevine, as a perennial woody plant and a major fruit crop cultivated by humans for millennia, is unique in several respects. First of all, it is infected by a large number of viruses that differ in genome structure, expression strategies and pathobiology. Secondly, grapevine viruses involved in major diseases comprises of multiple strains and genetic variants. Thirdly, grapevines grown for commercial purposes are commonly infected with a mixture of viruses and viral strains. Moreover, grapevines are susceptible to infection by other pathogens such as fungi, bacteria and nematodes. Such complex interactions make the elucidation of the individual impact of a given virus very difficult. However, when investigating the effects of a specific virus on the grapevine host, one must consider the pathobiome of the affected plant, i.e., the population of viruses and viral strains that co-exist in the infected grapevine.

Grapevine leafroll is a highly destructive disease that causes major economic losses to the grape and wine industries throughout the world. GLRD has a complex etiology as multiple viruses belonging to three genera of the family *Closteroviridae* have been associated with the disease. To further complicate the situation, large numbers of genetic variants are known to exist for each of these viruses. Furthermore, and as stated above, commercial grapevines are commonly infected by a multitude of distinct viruses. Numerous studies have attempted to unravel the molecular mechanisms on how GLRaVs interact with the grapevine host leading to the development of the disease symptoms typical of GLRD. These research efforts produced valuable information pertaining to the understanding of GLRD, its pathobiology and the development of disease symptoms. However, these studies are often limited in scope, targeting a single tissue type (either leaves or berries) collected at a specific growth stage. As a result, these studies often report inconsistent and sometimes even contradictory results. Other factors that might have resulted in discrepancies in findings from different studies may include the presence of other viruses, differences in the rootstocks used, viral load, as well as environmental conditions. 

Here, we attempt to propose a working model to explain how infection with GLRaVs leads to grapevine leafroll disease and its characteristic symptoms (Figure 4). We feel that a model, though rudimentary, can serve as a starting point towards the ultimate resolution of the pathobiology and disease development of this economically important disease complex. Because GLRaV-3 is considered the major agent of GLRD and as most of the studies that have been so far conducted focused on this virus, the model below mainly concerns GLRaV-3. It is likely that this model also applies to GLRaV-1 given the similarities between these two viruses in terms of genome architecture, expression strategies and pathobiology.

GLRaV-3, as a phloem-restricted virus, replicates in cells of the phloem tissue (sieve elements, companion cells and parenchyma cells) upon entry into the vine through vegetative propagation and insect vectors. As GLRaV-3 replicates over multiple years, it increases in titer and causes damage to the infected cells such as the destruction of mitochondria. These events eventually lead to blockage of sugar transport from leaves to the berries. Consequently, sucrose accumulates in source leaves, triggering the expression of various genes of the flavonoid pathway. In turn, this leads to the increased conversion of anthocyanidins to anthocyanins, producing the typical discoloration of leaf blades in dark-skinned grape cultivars. On the other hand, some of the berries do not receive sufficient sugar due to the blockage in sugar transport, resulting in reduced size and pigmentation. Furthermore, the blockage in sugar transport leads to repression of genes involved in photosynthesis and induces the degradation of chlorophyll. Therefore, white-skinned cultivars, which lack enzymes required for anthocyanin production, exhibit the typical yellowing symptoms due to the loss of chlorophyll. Moreover, and as suggested by Halldorson and Keller [109], the increased levels of soluble sugar in source leaves cause the turgor pressure to rise in mesophyll cells from absorbing excess amounts of water, leading to cell expansion. Because the palisade layer contains densely packed mesophyll cells, cell expansion of the palisade layer will force the downward curling of the leaf blade.

Many questions concerning the biology of GLRaVs and how each GLRaV interacts with the grapevine host leading to disease remain to be answered. To understand the process as a whole, both leaf and berry samples from different growth stages need to be analyzed in future research. Moreover, the effects of changes in gene expression at the transcription level need to be validated at the proteomic and metabolomic levels. Furthermore, the effects of infection by individual grapevine viruses of the *Closteroviridae* need to be elucidated. The only way to achieve this is through infection of clean grapevine that are free of all major viruses with infectious viral clones for each of these GLRaVs. To date, infectious viral clones for a strain of GLRaV-2 and two strains of GLRaV-3 have been constructed ([157,158], and our unpublished data). Few studies have been conducted on the frequency of mixed infection of GLRaVs. It remains an open question whether mixed infection of different GLRaVs would induce synergistic or antagonistic effects. In addition, basic research on viral replication, cytopathology and cell biology of viral infections as well as the functional characterization of genes unique to each GLRaV warrant investigation.

## Figures and Tables

**Figure 1 viruses-13-00593-f001:**
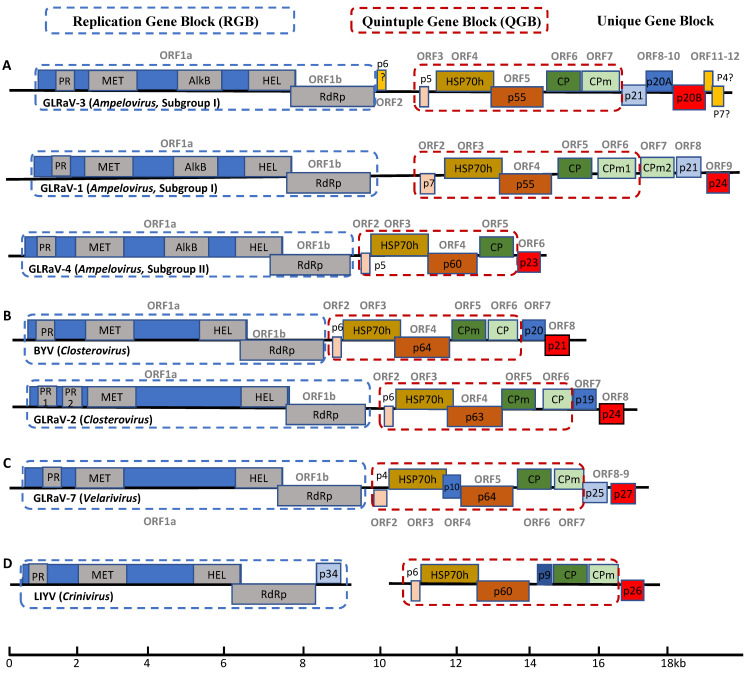
Genome organization of representative viruses of the family *Closteroviridae*. (**A**) viruses of the genus *Ampelovirus* including GLRaV-1 and GLRaV-3 (Subgroup I) and GLRaV-4 Subgroup II); (**B**) Viruses of the genus *Closterovirus* include BYV (the prototype member) and GLRaV-2; (**C**) GLRaV-7 is the prototype member of the genus *Velarivirus*; (**D**) the prototype of genus *Crinivirus*, LIYV. All viruses of the family *Closteroviridae* have a monopartite genome of single-stranded, positive sense RNA whereas members of the genus *Crinivirus* contain either bipartite or tripartite RNA genomes. All viruses of this family share a basic architectural design with the conserved replication gene block (RGB) and the quintuple gene block (QGB) with variable number of unique genes located toward the 3′ end of the viral genome. PR: papain-like protease; MET: methyl transferase; AlkB: alkylation B domain; HEL: RNA helicase; RdRP: RNA-dependent RNA polymerase; HSP70h: homolog of HSP70; CP: major capsid protein; CPm: minor capsid protein. Three ORFs in GLRaV-3 [ORF2 (p6), ORF11 (p4) and ORF12 (p7)] are questionable as it is unknown if any of them is actually translated. Note that two copies of CPm are encoded by GLRaV-1, whereas GLRaV-4 lacks the ORF for CPm.

**Figure 2 viruses-13-00593-f002:**
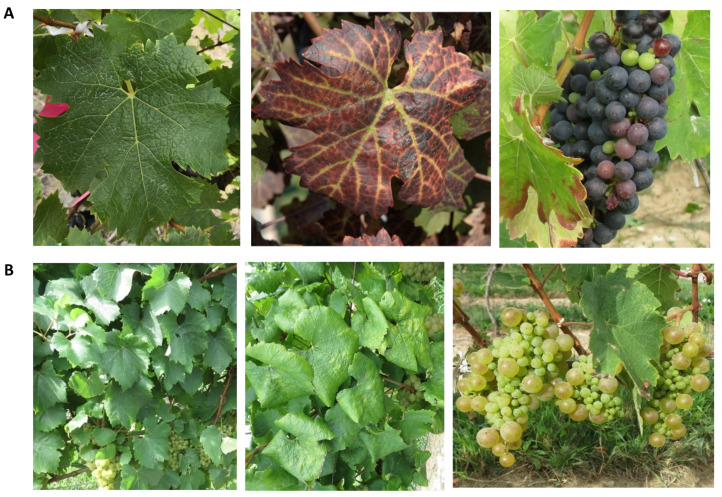
Symptomatology of grapevine cv. Cabernet franc (**A**) and cv. Chardonnay (**B**) infected with GLRaV-3. Left: leaf of vine not infected with GLRaV-3; middle: leaf of vine infected with GLRaV-3; right: berry cluster from GLRaV-3 infected plants. The typical leaf symptoms of GLRaV-3 infection, downward rolling of leaf margins and interveinal discoloration (reddening in dark-berried grapevines and chlorosis in white-skinned cultivars) are readily seen. Photos in B are from Shabanian et al. (2020) [36].

**Figure 3 viruses-13-00593-f003:**
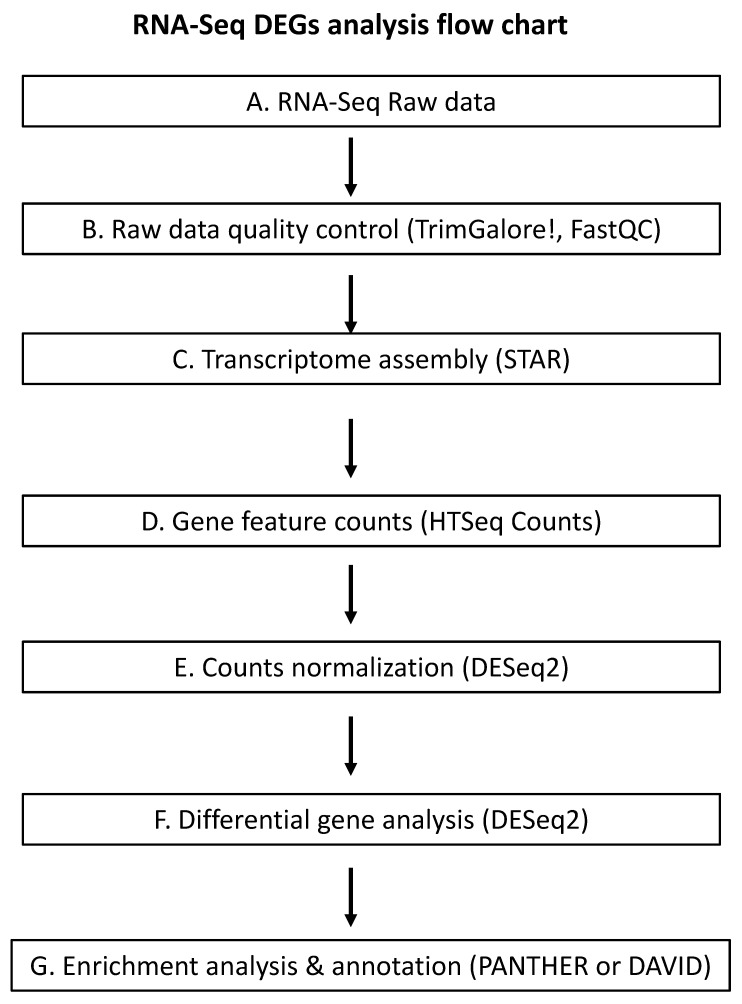
Typical pipeline for RNA-Seq data analysis for differential gene expression analysis with known reference genome consists of five steps, raw data quality control, transcriptome assembly, gene feature counts, counts normalization, differential gene analysis, functional annotation. Example of bioinformatics tools for each step is listed in bracket.

**Figure 4 viruses-13-00593-f004:**
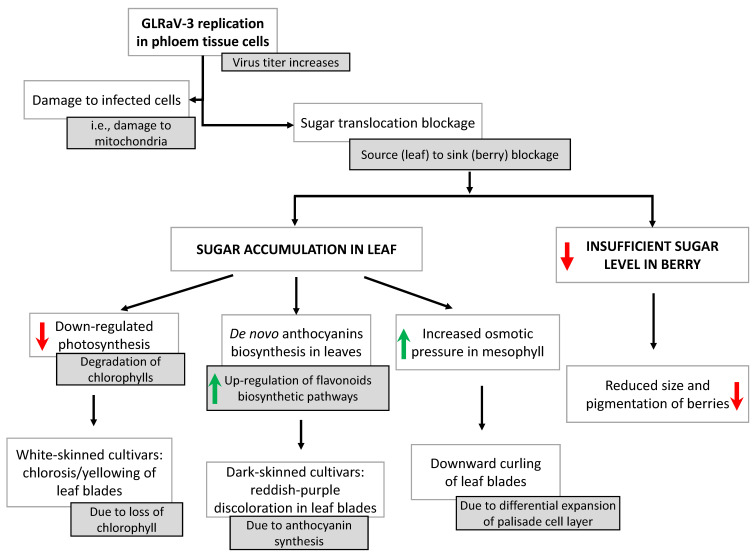
Schematic depiction of our working model on the molecular interactions between GLRaVs and the grapevine host that lead to GLRD and its characteristic symptoms. Red-arrows indicate up-regulation, green arrows indicate down-regulation.

**Table 1 viruses-13-00593-t001:** Summary of transmission routes of GLRaVs.

Virus	Virus Genus	Natural Transmission (Biological Vector)	Manual Transmission
GLRaV-1	*Ampelovirus* (Subgroup I)	Mealybugs and soft scale insects	Global exchange and propagation of virus-infected grapevine material
GLRaV-2	*Closterovirus*	Not identified	
GLRaV-3	*Ampelovirus* (Subgroup I)	Mealybugs and soft scale insects	
GLRaV-4	*Ampelovirus* (Subgroup II)	Mealybugs and soft scale insects	
GLRaV-7	*Velarivirus*	Parasitic dodder plants	

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
