# Peer review of "Probing into the Effects of Grapevine Leafroll-Associated Viruses on the Physiology, Fruit Quality and Gene Expression of Grapes"

_viruses, 2021, doi:10.3390/v13040593_

Round 1
Reviewer 1 Report
This review by Song et al. summarizes the information known about Grapevine leafroll virus, a collection of viruses from the Closteroviridae that cause substantial damage to grapevine. The review is very informative and I enjoyed learning more about this disease. I have made a number of comments directly on the manuscript to correct for grammar among other issues. I only list the important issues here that should be considered prior to publication. With these changes, this will be a very nice review of the field.
Major issues:
- The entire section on methodologies should be omitted as being unrelated to the topic. Methodologies used by researchers that are relevant to the results being summarized here should be included with the earlier statements. Readers are free to access detailed reviews on the methodologies, as they are not unique to the study of this disease.
- Figure 1 should have the ORF numbers designated. It was confusing to follow.
- Readers not familiar with these viruses are unaware of the sgRNAs that are present for these viruses and their unusual nature. I would include statements about this before discussing sgRNAs on line 168
- The authors included very limited information on how these viruses are transmitted, and a table on this would have been beneficial.
- There is little information on how frequently these viruses are found together or alone. Is there any incompatibility like for CTV?
- Rather than the section on methodologies, I would have appreciated a time line for how quickly these viruses are spreading and what is being done towards eradication. What are the possibilities for HIGS/VIGS approaches? What are the most beneficial control methods (for the vector)? Where in the world is this a major/minor problem and what does the future hold if no solution is found?
- Unless I missed it, the fact that these viruses are phloem-limited should be presented earlier.

Author Response
Dear Reviewer 1, please kindly find the attachment Cover Letter for our point-to-point response to your comments
Best

Reviewer 2 Report
Manuscript viruses-1146964 provides a brief but complete overview of the family closteroviridae, presenting the systematic of GLRD associated viruses in a correct manner.
This work offer a valid attempt to summarize the relevant literature dealing with the pathological impacts of GLRD associated viruses on the grapevine, also considering the molecular mechanisms underlying this interaction, which have yet to be fully elucidated.
In addition, the section dedicated to methodologies to study effects of GLRaVs on grapevine is well presented and the most recent finding are properly summarized, so as not to be too heavy for the reader. Both the advantages and the limitations of each methodology are correctly reported to my opinion. Although the section dedicated to RNAseq is rather long and detailed, it offers in my opinion a greater completeness of information, especially for those with no experience in this field.
This work manages to bring order among different studies with highly variable findings, which in most cases can be difficult to interpret.To my opinion, the whole paper is exhaustive, well written and organized; figures well support the text body.. For these reasons, I can recommend its publication in Viruses.
Minor suggestions:
- 1, Line 37: I suggest ‘by various Homoptera insects’ instead of ‘by various insects of Homoptera’
- 9, line 441: ‘in leaves of vines infected by GLRaV-3’ instead of ‘in leaves of vines infected GLRaV-3’
I would reorganize the Figure 2 caption as follows:
Symptomatology of grapevine cv. Cabernet franc (a) and cv. Chardonnay (b) infected with GLRaV-3. Left: leaf of vine not infected with GLRaV-3; middle: leaf of vine infected with GLRaV-3; right: berry cluster from GLRaV-3 infected plants. The typical leaf symptoms of GLRaV-3 infection, downward rolling of leaf margins and interveinal discoloration (reddening in dark-berried grapevines and chlorosis in white-skinned cultivars) are readily seen.
Author Response
Dear Reviewer 2, please kindly find the attachment Cover Letter, specifically, at page 7 of Cover Letter, for our point-to-point response to your comments
Best

Reviewer 3 Report
This review is divided in 5 sections. The first section presents an overview of the genome organisation and gene expression of the closteroviruses family followed by a section describing the main features of the grapevine leafroll disease and the 6 closteroviruses associated. The third section reports data about the impact of the leafroll disease on grapevine physiology and correlation with metabolic/genetic pathway impacted by the virus infection. The forth section is focused on methodologies used to study how GLRaV infections alter gene expression. This manuscript is ended by concluding remarks and proposes a model of plant/viruses interactions that explain the symptoms production.
This manuscript is well written, well-structured, and documented with recent data. This review is of interest for the scientific community working on those detrimental viruses for the wine industry throughout the world. The proposition of a working model is welcome and well summarize the knowledge acquired on grapevine/Closterovirus interaction. This review demonstrate that the authors has a good knowledge of this research area. My recommendation is to publish this manuscript after taking into account some suggestions/comments listed below.
1) I suggest summarizing the last four paragraphs of the section 4 because the content of those paragraphs is essentially technical and is not really specific to closterovirus or grapevine, two main topics of this revue. Also, It will reduce the length of this section that is quite long.
2) Figure 1 : As the authors specify the genome/gene lengths in the two first sections, I would suggest to add a kb scale bare at the bottom of the figure1. It will help to visualize the sizes of the genes/genomes described in the manuscript. Also, the black annotation/number in the red boxes are hidden by the red colour.
3) Figure 2 : Pictures resolution should be improved.
4) To illustrate the proposed working model in the last section, I suggest to add a schematic representation of the Grapevine/Closterovirus interaction to explain how viruses infection leads to grapevine leafroll disease.
5) In the references section, some years are in bold character and some are in normal character.
Author Response
Dear Reviewer 3, please kindly find the attachment Cover Letter, specifically, at page 8 of Cover Letter, for our point-to-point response to your comments.
Best
